# Profiling of Circulating microRNAs in Prostate Cancer Reveals Diagnostic Biomarker Potential

**DOI:** 10.3390/diagnostics10040188

**Published:** 2020-03-28

**Authors:** Jacob Fredsøe, Anne K. I. Rasmussen, Peter Mouritzen, Marianne T. Bjerre, Peter Østergren, Mikkel Fode, Michael Borre, Karina D. Sørensen

**Affiliations:** 1Department of Molecular Medicine, Aarhus University Hospital, 8200 Aarhus N, Denmark; jcf@clin.au.dk (J.F.); maribjrr@rm.dk (M.T.B.); 2Department of Clinical Medicine, Aarhus University, 8000 Aarhus C, Denmark; 3Exiqon A/S, Skelstedet 16, 2950 Vedbaek, Denmark; anne.karin.rasmussen@gmail.com (A.K.I.R.); mouritzenpeter0@gmail.com (P.M.); 4Department of Urology, Aarhus University Hospital, 8200 Aarhus N, Denmark; 5Department of Urology, Herlev and Gentofte Hospital, 2900 Hellerup, Denmark; peter.busch.oestergren@regionh.dk (P.Ø.); mikkelfode@gmail.com (M.F.)

**Keywords:** prostate cancer, biomarker, diagnosis, microRNA, plasma, bCaP

## Abstract

Early detection of prostate cancer (PC) is paramount as localized disease is generally curable, while metastatic PC is generally incurable. There is a need for improved, minimally invasive biomarkers as current diagnostic tools are inaccurate, leading to extensive overtreatment while still missing some clinically significant cancers. Consequently, we profiled the expression levels of 92 selected microRNAs by RT-qPCR in plasma samples from 753 patients, representing multiple stages of PC and non-cancer controls. First, we compared plasma miRNA levels in patients with benign prostatic hyperplasia (BPH) or localized prostate cancer (LPC), versus advanced prostate cancer (APC). We identified several dysregulated microRNAs with a large overlap of 59 up/down-regulated microRNAs between BPH versus APC and LPC versus APC. Besides identifying several novel PC-associated dysregulated microRNAs in plasma, we confirmed the previously reported upregulation of miR-375 and downregulation of miR-146a-5p. Next, by randomly splitting our dataset into a training and test set, we identified and successfully validated a novel four microRNA diagnostic ratio model, termed *bCaP* (miR-375*miR-33a-5p/miR-16-5p*miR-409-3p). Combined in a model with prostate specific antigen (PSA), digital rectal examination status, and age, *bCaP* predicted the outcomes of transrectal ultrasound (TRUS)-guided biopsies (negative vs. positive) with greater accuracy than PSA alone (Training: area under the curve (AUC), model = 0.84; AUC, PSA = 0.63. Test set: AUC, model = 0.67; AUC, PSA = 0.56). It may be possible in the future to use this simple and minimally invasive *bCaP* test in combination with existing clinical parameters for a more accurate selection of patients for prostate biopsy.

## 1. Introduction

With more than one million cases diagnosed annually, prostate cancer (PC) is one of the most common cancer types in males in the Western world [1]. Localized prostate cancer (LPC) is generally curable by surgery or radiation therapy with a 5-year survival rate of close to 100% while metastatic prostate cancer (MPC) is generally incurable and has a 5-year survival rate below 40% [2,3]. However, many LPCs will not progress to an aggressive state, and even if left untreated would not give rise to any symptoms in the patient’s normal lifespan. Consequently, early and accurate diagnosis is essential for long-term patient survival and for preventing overtreatment.

The primary diagnosis of PC is typically performed by transrectal ultrasound guided biopsies (TRUSbx) of the prostate, prompted by increased serum levels of prostate specific antigen (PSA) and/or suspicious findings on digital rectal examination (DRE). However, this approach is suboptimal, resulting in up to 75% of all initial TRUS-biopsies being negative [4], while still missing 25% of clinically significant cancers [5]. Furthermore, DRE has a low sensitivity and is dependent on the experience of the examiner [6,7]. In recent years, multiparametric MRI (mpMRI) with MRI-targeted biopsies has been proven to be an improvement for PC detection over standard systematic TRUSbx [8,9,10]. This is also reflected in the recently updated European guidelines for PC [11], which now recommend mpMRI before systematic biopsies. However, mpMRI is a limited resource, unavailable at some centers, and still misses some PCs that may be detected by the random, but systematic TRUSbx approach [12]. Taken together, there is a pressing need for development of more precise and less invasive biomarkers that can provide guidance for a more personalized diagnostic approach.

In recent years, liquid biopsies (e.g., blood, urine, and other bodily fluids) have gained attention for investigating circulating tumor DNA, RNA, or microRNAs (miRNAs) in minimally invasive tests for diagnosis, prognosis, recurrence, and disease monitoring [13,14,15,16]. miRNAs are short non-coding RNAs approximately 22 nucleotides in length, which post-transcriptionally regulate gene expression of up to 60% of all human mRNAs and often play a significant role in oncogenesis [13,17,18]. Dysregulation of miRNA expression in PC has previously been demonstrated in both tumor tissue samples and liquid biopsies [19,20,21,22,23,24]. Coupled with the high stability of miRNAs in biofluids [25], this makes them particularly appealing targets as minimally invasive biomarkers.

In the current study, we profiled the expression levels of 92 individual miRNAs in plasma samples from more than 750 patients representing several different stages of the diagnostic process of PC, including patients undergoing initial TRUSbx, or diagnosed with either benign prostatic hyperplasia (BPH), clinically localized prostate cancer (LPC), or advanced prostate cancer (APC). Here, we report several dysregulated miRNAs between the different patient groups. Furthermore, we train and successfully validate a novel miRNA-based ratio model (termed *bCaP*) for predicting the outcomes of TRUSbx with greater accuracy than PSA.

## 2. Materials and Methods

### 2.1. Cohort Characteristic

We analyzed plasma samples from four distinct patient groups (see Table 1 for an overview of clinicopathological characteristics). Group 1 contained patients with benign prostatic hyperplasia (BPH) undergoing trans-urethral resection of the prostate (TURP). Group 2 consisted of patients with histologically verified clinically localized PC (LPC) treated by curatively intended radical prostatectomy (RP). Group 3 consisted of patients with hormone naïve advanced PC (APC) with an indication for life-long androgen deprivation therapy (ADT); i.e., having verified metastatic PC at time of diagnosis or high-risk features excluding curative treatment (e.g., PSA > 75 ng/mL and/or T4 disease on mpMRI). Finally, group 4 consisted of patients with suspicion of PC undergoing initial TRUS-guided biopsy (TRUSbx, >96% with at least 10 needles), and was further subdivided into those with malignant findings in the needles and those where all needles were histopathologically verified to be free of cancer.

Plasma samples from groups 1 (BPH), 2 (LPC), and 4 (TRUSbx) were collected at the Department of Urology, Aarhus University Hospital, Denmark (2001–2018) or Aleris Hamlet Hospital, Aarhus, Denmark (2017–2018). Plasma samples from group 3 (APC) were collected at the Department of Urology, Herlev and Gentofte Hospital, Herlev, Denmark (2013–2015). For groups 1, 2, 3, and 4, blood samples were drawn into tubes containing EDTA just prior to TURP, RP, start of ADT, or TRUSbx, respectively. In all cases, whole blood was spun at 2000× *g* (group 1, 2, and 4) or 3020× *g* (group 3) for 10 min at 4 °C before the plasma was transferred into fresh cryotubes and stored at −80 °C until use. The study was approved by the regional scientific ethics committee (journal no. 1-10-72-34-16) and notification was given to the Danish Data Protection Agency (journal no. 2013-41-2041). Written informed consent was obtained from all patients.

We measured the expression levels of 92 miRNAs in plasma from all patient groups using a miRNA reverse transcriptase–polymerase chain reaction platform from Exiqon (Vedbaek, Denmark). Briefly, total RNA was extracted from 200 µL serum using the miRCURY™ RNA isolation kit, biofluids (Exiqon, Vedbaek, Denmark) according to the manufacturer’s description and stored at −80 °C. Next, 10 µL RNA was reverse transcribed using the miRCURY LNA™ Universal RT microRNA PCR, Polyadenylation, and cDNA synthesis kit (Exiqon) followed by expression analysis by PCR using universal RT Pick-&-Mix microRNA PCR panels, V4.R (Exiqon, Vedbaek, Denmark), according to the manufacturer’s description and as described in detail previously [14,21].

### 2.2. Statistical Analysis

All statistical analyses were conducted in R (version 3.5.1) [26] using R studio, version 1.1.383. Initially, batch effects were corrected using the limma package [27] followed by normalization of Cq values to the mean of the most stably expressed pair of miRNAs (miR-23a-3p and miR-93-5p) as determined by the Normfinder [28] algorithm. Accordingly, each single miRNA was normalized as ΔCq = mean(Cq_miR-23a-3p_,Cq_miR-93b-5p_) − Cq_miRNA_. The Wilcoxon rank-sum test was used to test for differences in miRNA levels between sample groups and *p* values were adjusted for multiple testing using the Benjamini–Hochberg (BH) approach. Fold changes were calculated as ΔΔCq = mean (ΔCq_group1_) − mean (ΔCq_group2_) and converted on a log2 scale. Diagnostic potential of PSA, single miRNAs, and *bCaP* was evaluated by receiver operating characteristic (ROC) curve analysis using the pROC package [29]. Cox regression analyses were performed using the survival package [30]. For recurrence-free survival analysis, the endpoint was biochemical recurrence (BCR) (PSA ≥ 0.2 ng/mL). Patients who had not experienced BCR were censored at their last normal PSA test.

CAPRA-S score [31] was calculated as a sum of the following: PSA at RP [6,7,8,9,10] = +1, (10,20] = +2, >20 = +3; prostatectomy Gleason Score 3 + 4 = +1, 4 + 3 = +2, ≥8 = +3; positive for surgical margin (SM) = +2; positive for seminal vesicle invasion (SVI) = +2; positive for extracapsular extension (ECE) = +1; positive for lymph node invasion (LNI) = +1. CAPRA-S risk groups were then defined as: [0,2] = low risk; [3,5] = intermediate risk; [6,12] = high risk.

For creation of the *bCaP* model, we randomly divided the dataset into a training (66%) and a test set (34%). Next, we used the top 20 most differentially expressed miRNAs between patients with a benign or malignant biopsy outcome in the training set, ranked by descending ROC area under the curve (AUC), while also being expressed in more than 90% of all TRUSbx samples in the training set (*n* models = 32,680). In the training set, each model was then evaluated by ROC AUC for its ability to differentiate between samples from patients with a benign biopsy outcome and patients with a malignant biopsy outcome. The top model in the training set (*bCaP*, miR-375*miR-33a-5p/miR-16-5p * miR-409-3p) was further evaluated in samples from BPH, LPC, and APC patients. Logistic regression models, combining serum PSA, *bCaP*, DRE, and age into a merged model (either PSA + *bCaP* or PSA + *bCaP* + DRE + age; Appendix A) were trained by use of the stats package [26] using the training set exclusively.

## 3. Results

### 3.1. Dysregulated miRNAs in Plasma

In order to investigate the potential of miRNA in plasma as diagnostic biomarker candidates for PC, we analyzed the expression levels of 92 unique miRNAs in plasma samples from 144 patients with benign prostatic hyperplasia (BPH), 407 patients with clinically localized prostate cancer (LPC), and 57 patients with hormone naïve advanced prostate cancer (APC) (Table 1). Of the latter, 51 patients were MPC (confirmed by imaging) while the remaining six had high risk features suggestive of undetectable dissemination rendering them unfit for curative treatment.

Initially, we identified 44 upregulated and 18 downregulated miRNAs in plasma samples from APC as compared to samples from BPH patients (*p* < 0.05, Wilcoxon rank sum test, Benjamini–Hochberg (BH) adjusted for multiple testing; Appendix A). Similarly, we found 45 upregulated and 28 downregulated miRNAs in plasma samples from APC as compared to LPC patients (BH adjusted *p* < 0.05, Appendix A). Of these, 59 miRNAs (80.8%) were also significantly up/down-regulated between BPH and APC samples.

Relative to BPH, the top five most downregulated miRNAs in APC were miR-146a-5p, miR-376c-3p, miR-410-3p, miR-154-5p, and miR-130a-3p, which were also found to be downregulated in APC relative to LPC (Table 2). Similarly, the top five most upregulated miRNAs in APC were miR-375, miR-26a-5p, miR-142-3p, miR-451a, and miR-215-5p, which were also upregulated in APC relative to LPC (Table 2). Thus, in addition to identifying several novel dysregulated miRNAs in plasma samples from PC patients, we were able to confirm a significant upregulation of the established PC-associated oncomir miR-375 [32,33,34,35], and a significant downregulation of the tumor suppressor miR-146a-5p [36,37,38].

### 3.2. Plasma miRNAs Associated with Prostate Cancer Aggressiveness

Next, using plasma samples from 407 LPC patients (all treated by RP), we investigated possible associations between miRNA levels in plasma and key clinical parameters associated with PC aggressiveness. Accordingly, we compared miRNA expression levels in patients with low/high PSA levels (<= 10 versus > 10 ng/mL), low/high pathological T stage (<= T2 versus >= T3), and low/high post-operative Gleason grade group (<=2 versus >2), tested correlations to CAPRA-S score [31], and performed BCR-free survival analyses by univariate Cox regression. After adjustment for multiple testing, there remained almost no significant associations, with the exception of four miRNAs that were upregulated (miR-122-5p, miR-29a-3p, miR-23-3p, and miR-375) and five miRNAs that were downregulated (miR-140-5p, miR-376c-3p, miR-93-5p, miR-376a-3p, miR-20a-5p) in patients with high PSA (BH adjusted *p* < 0.05; Appendix A).

In summary, despite investigating a representative and relatively large RP patient set, we found little association of the 92 miRNAs investigated and parameters normally associated with poor prognosis of PC. Most notably, we observed increased expression levels (1.49 fold) of circulating miR-375 in LPC patients with high PSA levels (>10 ng/l) relative to patients with low PSA.

### 3.3. Diagnostic Potential of Plasma miRNAs

Next, we investigated the diagnostic biomarker potential of miRNAs in plasma samples taken before initial TRUSbx, using a consecutive cohort of 145 patients (63 with benign outcomes and 82 with malignant outcomes, Table 1). We found four dysregulated (all upregulated) miRNAs in patients with malignant biopsy outcomes (Wilcoxon test, *p* < 0.05; miRNA-375, miRNA-99a-5p, miRNA-19a-3p, and miRNA-16-2-3p; Appendix A), but none remained significant after adjusting for multiple testing. Out of these four miRNAs, the expression levels of miRNA-375, miRNA-19a-3p, and miRNA-16-2-3p were also upregulated in samples from APC compared to LPC patients (adjusted *p* < 0.05, Appendix A), while miRNA-99a-5p levels were unchanged.

The lack of any significant individual discriminatory miRNAs for TRUSbx outcome prompted us to attempt to combine miRNAs in ratio models. Ratio models have the distinct advantage of being simple and, by design, circumvent the need for addition normalization. Accordingly, we randomly split all patient samples (with no regard to which patient group the samples belonged to) into a training set (66% of all samples) and a test set (34% of all samples). Using TRUSbx samples from the training set exclusively (41 benign and 62 malignant outcome), we generated all unique two, three, and four-miRNA ratio models from the top 20 most differentially (as evaluated by descending ROC AUC) and consistently expressed (>90% of all training samples) miRNAs between benign and malignant outcomes of TRUSbx (*n* models = 32,680).

The best model in the training set was a 4-miRNA model (miR-375 * miR-33a-5p/miR-16-5p * miR-409-3p; termed *bCaP* from now on) with an AUC of 0.73 between benign and malignant outcomes (Figure 1a and Figure 2a), where a higher *bCaP* score signified a malignant outcome. This was an increase in AUC of 0.11 from the best single-miRNA (miR-375, AUC = 0.62, data not shown) and a greater AUC than for PSA (AUC = 0.63, Δ = 0.10, Figure 1b and Figure 2b). Next, we attempted to validate *bCaP* in the test set, and found that *bCaP* differentiated positive vs. negative biopsy outcomes with an AUC of 0.68, which again was greater than the AUC for PSA (AUC = 0.56, Δ = 0.12, Figure 2b). Finally, we tested *bCaP* in patient samples from the BPH (*n* = 89/55 for training/test sets), LPC (*n* = 265/142), and APC (*n* = 39/18) groups and found that *bCaP* score increased gradually with advancing disease states (i.e., from BPH to LPC to APC, Figure 1a). Importantly, there was no significant difference in *bCaP* score between samples from patients with benign TRUSbx and BPH patients, nor between patients with malignant TRUSbx and LPC patients in either the training or the test set (*p* > 0.05, Wilcoxon Test, Figure 1a). In contrast, *bCaP* scores were significantly (*p* < 0.001, Wilcoxon test, Figure 1a) increased in samples from malignant biopsies, LPC, and APC patients relative to samples from benign TRUSbx patients in the training set. In the test set, *bCaP* scores were significantly higher in APC samples (Wilcoxon test, *p* < 0.05, Figure 1a), but did not reach significance in malignant biopsies or LPC (*p* = 0.07 and 0.06, respectively) relative to samples from benign biopsy patients.

While *bCaP* had higher AUC than PSA for predicting TRUSbx outcomes, PSA had better accuracy for advanced stages of PC (Figure 1b). Consequently, we created a logistic regression model, using the training set exclusively, that combined *bCaP* and PSA, or *bCaP*, PSA, DRE, and age. Using this full model resulted in an AUC of 0.84, which was greater than for *bCaP* (AUC = 0.73) or PSA (AUC = 0.63) alone in the training set (Figure 2a), and had a comparable AUC (0.67) to *bCaP* (AUC = 0.68) alone but higher AUC than PSA alone (AUC = 0.56) when used in the test set (Figure 2b). In conclusion, our results suggest that it is possible to develop a simple non-invasive diagnostic classifier that could aid in the primary diagnosis of PC.

## 4. Discussion

We profiled the expression levels of 92 individual miRNAs using plasma samples from 753 patients in four different patient type groups (BPH, PC, APC, and TRUSbx). We identified multiple significantly dysregulated miRNAs between the groups, but few miRNAs were significantly associated with PC aggressiveness. Furthermore, we trained and validated a novel four-miRNA ratio model (*bCaP*) that could predict the outcome of TRUSbx with a higher accuracy than PSA. To the best of our knowledge, this constitutes the most comprehensive study of circulating miRNAs in plasma using samples from multiple stages of PC diagnosis.

The most upregulated single miRNA in samples from patients with APC or malignant biopsies was miR-375 (which is also included in the *bCaP* model; miR-375 * miR-33a-5p/miR-16-5p * miR-409-3p), consistent with the growing body of evidence for the oncogenic role of this miRNA in PC [19,32,33,34,35]. Out of the three other miRNAs in *bCaP*, only miR-16-5p was significantly upregulated in samples from patients with APC, relative to that of samples from patients with BPH or LPC. miR-16-5 has previously displayed elevated expression levels in plasma from LPC patients compared to healthy controls [22,39] as well as elevated expression levels in serum of PC mouse models with bone metastasis [40]. We found no significant changes in expression levels in plasma between samples from BPH or LPC versus APC of either miR-33a-5p or miR-409-3p, indicating that these two miRNAs may simply function as normalization genes in our *bCaP* model. Still, studies using PC tissue analyses have implicated miR-33a-5p [41] and miR-409-3p [42] to play a role in bone-metastasis and epithelial-to-mesenchymal transition, respectively. However, a large-scale plasma miRNA profiling study [22], in line with our results, did not detect any difference in plasma levels between miR-33a-5p and miR-409-3p between patients with benign TRUSbx outcomes and patients with LPC. Thus, miRNA expression levels observed in tumor tissue may not fully reflect the expression levels observed in circulation.

Ratio models, such as *bCaP*, are attractive, as they require no additional normalization by design. Potentially, this may ease future implementation into clinical use as it requires only a simple and straightforward laboratory setup (real-time PCR). When *bCaP* was combined with PSA, DRE, and age, the resulting model had increased accuracy over PSA alone (Figure 2) in both the training and test sets. Therefore, our results suggest that *bCaP* could be used in combination with existing clinical parameters to select which patients should undergo TRUSbx, although further large-scale clinical testing is required.

In a Japanese study by Urabe et al. [22], the authors profiled 2588 miRNAs in >800 plasma samples from non-cancer patients and patients with LPC and found a two-miRNA model consisting of miRNA-17-3p (AUC > 0.91) and miR-1185-2-3p (AUC > 0.92) which, when combined, could detect PC with very high accuracy (AUC > 0.95). In the Urabe dataset, *bCaP* could not significantly differentiate between non-cancer and LPC patients (AUC = 0.54). However, when tested in our dataset, circulating miR-17-3p was not significantly differentially expressed between BPH and LPC patients (AUC = 0.52; miR-1185-2-3p was not measured in our study). Consequently, it is possible that differences in ethnicity, sample preparation, handling, and miRNA measurement procedure between these two studies differed too much to make direct comparisons.

The lack of a true independent validation cohort is a limitation of the current study. However, our approach allowed for greater consistency in sample handling to reduce potential batch effects. Furthermore, while the TRUSbx group was rather limited in numbers compared to the LPC group, it was possible to train a model in a more clinically relevant setting. As the biopsy patients in our cohort all were previously biopsy naïve, there is a possibility that some patients with benign outcomes had occult cancer that was missed by initial TRUSbx. Furthermore, as mpMRI and targeted biopsy is now preferred before TRUSbx [11], future studies should include larger cohorts and patients referred for prostate mpMRI. Finally, we did not characterize the origin of the miRNAs investigated in the current study and cannot know if expression changes observed originated from prostate (cancer) tissue.

In conclusion, a comprehensive analysis of miRNA expression levels in plasma from different stages of prostate cancer identified a 4-miRNA ratio model (*bCaP*) that may potentially be used to better select patients for prostate biopsy in the future. Further validation is warranted.

## Figures and Tables

**Figure 1 diagnostics-10-00188-f001:**
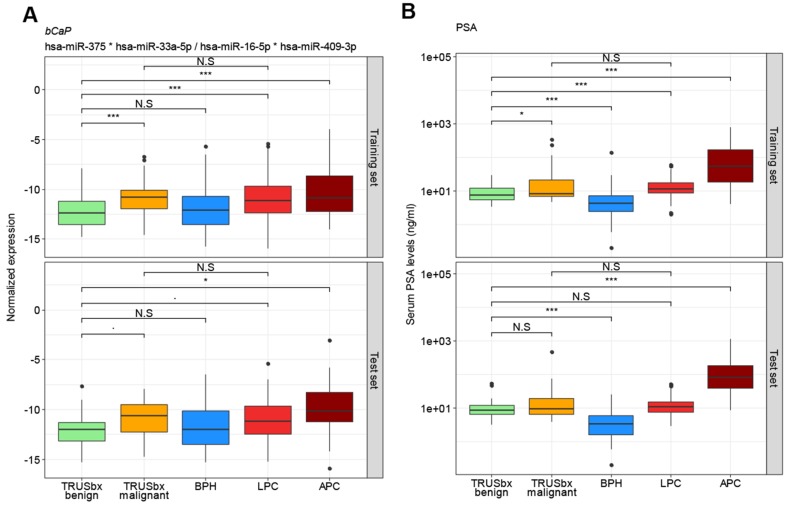
Box plot analysis for the four-miRNA diagnostic model *bCaP* (miR-375*miR-33a-5p/miR-16-5p*miR-409-3p) (**A**) and PSA (**B**) in the training (top) or test (bottom) sets. Boxes represent the first and third quartiles. The median is shown as a horizontal line and dots indicate outliers. *p* values indicate the difference in median, by Wilcoxon rank sum test. N.S = not significant (*p* > 0.05), ∙ = *p* < 0.1; * = *p* < 0.05; *** = *p* < 0.001; BPH (blue): benign prostatic hyperplasia; LPC (red): localized prostate cancer; APC (dark red: advanced prostate cancer; TRUSbx (green and orange: transrectal ultrasound (TRUS) guided biopsy; PSA: prostate specific antigen.

**Figure 2 diagnostics-10-00188-f002:**
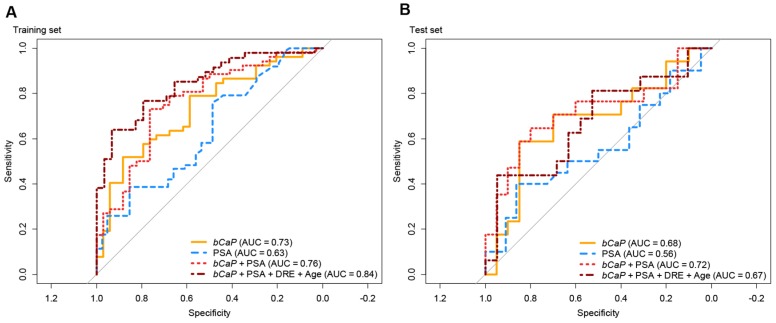
Receiver operating characteristic (ROC) curve analysis in training (**A**) and test (**B**) sets for predicting malignant outcome of TRUSbx. Orange: *pCaP*; blue: PSA; red: logistic regression model of *bCaP* + PSA; dark red: logistic regression model of *bCaP* + PSA + DRE + Age; PSA: prostate specific antigen; DRE: digital rectal examination.

**Table 1 diagnostics-10-00188-t001:** Summary of clinicopathological characteristics of patient cohorts.

	BPH	LPC	APC	TRUSbx Benign	TRUSbx Malignant
**Number of Samples**	*n* = 144	*n* = 407	*n* = 57	*n* = 63	*n* = 82
**Median Age (range)**	70 (46–87)	64 (36–77)	78 (47–86)	66 (43–80)	68 (43–80)
**Serum PSA levels, *n* (%)**					
≤10 ng/mL	114 (79.2%)	155 (38.1%)	5 (8.8%)	41 (65.1%)	43 (52.4%)
>10 ng/mL	20 (13.9%)	252 (61.9%)	55 (96.5%)	22 (34.9%)	39 (47.6%)
Unknown	10 (6.9%)	0 (0%)	0 (0%)	0 (0%)	0 (0%)
**Median PSA, ng/mL (range)**	4.1 (.2–141)	11.40 (2–61)	78 (4.1–1147)	8.4 (3.3–54.7)	8.6 (4–466)
**Pre-biopsy DRE status, *n* (%)**					
Positive	NA	NA	NA	14 (22.2%)	45 (54.9%)
Negative	NA	NA	NA	42 (66.7%)	30 (36.6%)
Unknown	NA	NA	NA	7 (11.1%)	7 (8.5%)
**T-stage, *n* (%) ***					
T1	NA	0 (0%)	3 (5.3%)	NA	26 (31.7%)
T2	NA	276 (67.8%)	12 (21.1%)	NA	17 (20.7%)
T3	NA	131 (32.2%)	35 (61.4%)	NA	36 (43.9%)
T4	NA	0 (0%)	5 (8.8%)	NA	0 (0%)
Unknown	NA	0 (0%)	2 (3.5%)	NA	3 (3.7%)
**Gleason Grade Group, *n* (%) ***					
1	NA	149 (36.6%)	2 (3.5%)	NA	21 (25.6%)
2	NA	188 (46.2%)	6 (10.5%)	NA	23 (28%)
3	NA	4 (1%)	8 (14%)	NA	5 (6.1%)
4	NA	52 (12.8%)	15 (26.3%)	NA	15 (18.3%)
5	NA	14 (3.4%)	24 (42.1%)	NA	18 (22%)
Unknown	NA	0 (0%)	2 (3.5%)	NA	0 (0%)
**Surgical margin status, *n* (%)**					
Negative	NA	282 (69.3%)	NA	NA	NA
Positive	NA	122 (30%)	NA	NA	NA
Unknown	NA	3 (0.7%)	NA	NA	NA

BPH: benign prostatic hyperplasia; LP: localized prostate cancer; AP: advanced prostate cancer; TRUSb: transrectal ultrasound (TRUS) guided biopsy; PS: prostate specific antigen; DR: digital rectal examination. * All LPC patients were treated by radical prostatectomy (RP), and T-stage and Gleason Grade Group were evaluated on the RP specimen. All APC were patients with verified metastatic PC at the time of diagnosis or high-risk features excluding curative treatment and consequently T-stage and Gleason Grade Group were from the time of diagnosis. NA: not available/applicable.

**Table 2 diagnostics-10-00188-t002:** Top five miRNAs upregulated and downregulated in APC relative to BPH (left) or LPC (right), ordered by fold change from BPH vs. APC. Benjamin–Hochberg was used to correct *p* values for multiple testing.

**Downregulated miRNAs in APC**	**Fold Change BPH vs. APC**	**BH Corrected *p* Value**	**Fold Change LPC vs. APC**	**BH Corrected *p* Value**
hsa-miR-146a-5p	−1.73	1.61 × 10^−10^	−1.82	2.45 × 10^−16^
hsa-miR-376c-3p	−1.64	1.66 × 10^−3^	−1.71	8.12 × 10^−5^
hsa-miR-410-3p	−1.59	1.66 × 10^−3^	−1.65	9.64 × 10^−5^
hsa-miR-154-5p	−1.51	7.89 × 10^−3^	−1.69	1.01 × 10^−4^
hsa-miR-130a-3p	−1.48	2.19 × 10^−5^	−1.37	8.93 × 10^−6^
**Upregulated miRNAs in APC**	**Fold Change BPH vs. APC**	**BH Corrected *p* value**	**Fold Change LPC vs. APC**	**BH Corrected *p* value**
hsa-miR-375	3.70	3.44 × 10^−6^	3.27	3.56 × 10^−6^
hsa-miR-26a-5p	1.89	2.51 × 10^−10^	2.55	2.52 × 10^−19^
hsa-miR-142-3p	1.89	6.39 × 10^−8^	2.76	1.06 × 10^−16^
hsa-miR-451a	1.84	1.21 × 10^−4^	3.33	1.76 × 10^−16^
hsa-miR-215-5p	1.75	2.36 × 10^−5^	1.82	2.50 × 10^−7^

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
