# Peer review of "Profiling of Circulating microRNAs in Prostate Cancer Reveals Diagnostic Biomarker Potential"

_diagnostics, 2020, doi:10.3390/diagnostics10040188_

Round 1

Reviewer 1 Report

The manuscript by Fredsøe et al., is a very interesting and comprehensive paper which describes the analysis of the expression levels of 92 specific microRNAs in plasma samples from 753 patients, representing multiple stages of prostate cancer and non-cancer samples. The authors define four microRNA diagnostic ratio model (bCaP) and compare their results with other classical predictive analysis (PSA, digital rectal examination…). Finally, they conclude that, basing on their data, bCaP test in combination other clinical parameters should be a simple and no-invasive test for a more accurate selection of patients for prostate biopsy.

The manuscript is very well written, and the authors have produced many positive and welcome outcomes. The manuscript could represent an important contribution in the field, but before acceptance some (minor) modifications are needed:

- Chapter 3.4 “Figures, Tables and Schemes” is no necessary. Tables are in pages 3-4-5 (and supplementary). There are not Schemes. Finally, figures should be placed immediately after being mentioned in the text.

- Line 300: I can not understand because the authors wrote two times their “fundings”

- References: Please check ref 11 and 30

Author Response

Please see our response to this reviewer in red below

The manuscript by Fredsøe et al., is a very interesting and comprehensive paper which describes the analysis of the expression levels of 92 specific microRNAs in plasma samples from 753 patients, representing multiple stages of prostate cancer and non-cancer samples. The authors define four microRNA diagnostic ratio model (bCaP) and compare their results with other classical predictive analysis (PSA, digital rectal examination…). Finally, they conclude that, basing on their data, bCaP test in combination other clinical parameters should be a simple and no-invasive test for a more accurate selection of patients for prostate biopsy.

The manuscript is very well written, and the authors have produced many positive and welcome outcomes. The manuscript could represent an important contribution in the field, but before acceptance some (minor) modifications are needed:

- Chapter 3.4 “Figures, Tables and Schemes” is no necessary. Tables are in pages 3-4-5 (and supplementary). There are not Schemes. Finally, figures should be placed immediately after being mentioned in the text.

We have now removed the section in question and moved the figures to the place immediately after being mentioned in the text.

- Line 300: I can not understand because the authors wrote two times their “fundings”

We have now removed the second “funding” heading.

- References: Please check ref 11 and 30

We have now corrected these (reference #30 is #42 in the revised manuscript).

Reviewer 2 Report

In this manscript, the possibility of application of circulating microRNAs as a biomarker of prostate cancer has been evaluated using samples from 753 patients in different stages of prostate cancer. Authors displayed superiority of prognostic value of 4 microRNA named as bCaP over PSA test. 

In my review, this is an interesting evidence of feasibility of application of microRNAs in prostate cancer screening. Also, the topic of this manscript is fully a relevant to the scope of Diagnostics. Therefore, I do recommend this submission for publication after below revision:

The manscript should be proofread for proper usage of professional medical terminology. For example, one first line of Abstract, incurable/curable could be replace with relevant survival data. Also, the intro should be relevant to the objectives of the paper. 

Author Response

Please see our response to this reviewer in red below

In this manscript, the possibility of application of circulating microRNAs as a biomarker of prostate cancer has been evaluated using samples from 753 patients in different stages of prostate cancer. Authors displayed superiority of prognostic value of 4 microRNA named as bCaP over PSA test. 

In my review, this is an interesting evidence of feasibility of application of microRNAs in prostate cancer screening. Also, the topic of this manscript is fully a relevant to the scope of Diagnostics. Therefore, I do recommend this submission for publication after below revision:

The manscript should be proofread for proper usage of professional medical terminology. For example, one first line of Abstract, incurable/curable could be replace with relevant survival data. Also, the intro should be relevant to the objectives of the paper. 

We have now changed the wording in the first line of the abstract to “generally curable” and “generally incurable”. In the first paragraph of the introduction, we already include the survival data. We have proofread the manuscript again, and found no need to make further changes as the language and terminology we use is common in the field.

We believe the introduction is already relevant to the objective of the paper, which is to investigate the biomarker potential of circulating microRNA for prostate cancer. Accordingly, the introduction covers prostate cancer in general, why there is a need for better biomarkers, microRNA in general and their role in cancer. Consequently, we have made no changes to the introduction.